# Impaired humoral immunity to BQ.1.1 in convalescent and vaccinated patients

Felix Dewald[1], Martin Pirkl [ID][1], Martha Paluschinski [ID][2], Joachim Kühn[3], Carina Elsner [ID][4], Bianca Schulte [ID][5,6], Jacqueline Knüfer[1], Elvin Ahmadov[1], Maike Schlotz[1], Göksu Oral[1], Michael Bernhard[7], Mark Michael[7], Maura Luxenburger[2], Marcel Andrée[2], Marc Tim Hennies[3], Wali Hafezi[3], Marlin Maybrit Müller[3], Philipp Kümpers[8], Joachim Risse [ID][9], Clemens Kill[9], Randi Katrin Manegold[9], Ute von Frantzki[9], Enrico Richter[5,6], Dorian Emmert[5], Werner O. Monzon-Posadas[10], Ingo Gräff[11], Monika Kogej[11], Antonia Büning[5], Maximilian Baum[5], Finn Teipel [ID][1], Babak Mochtarzadeh[1], Martin Wolff[1], Henning Gruell [ID][1], Veronica Di Cristanziano [ID][1], Volker Burst[12,13], Hendrik Streeck[5,6], Ulf Dittmer[4], Stephan Ludwig [ID][3], Jörg Timm[2] & Florian Klein [ID][1,6,14] ✉

Determining SARS-CoV-2 immunity is critical to assess COVID-19 risk and the need for prevention and mitigation strategies. We measured SARS-CoV-2 Spike/Nucleocapsid seroprevalence and serum neutralizing activity against Wu01, BA.4/5 and BQ.1.1 in a convenience sample of 1,411 patients receiving medical treatment in the emergency departments of five university hospitals in North Rhine-Westphalia, Germany, in August/September 2022. 62% reported underlying medical conditions and 67.7% were vaccinated according to German COVID-19 vaccination recommendations (13.9% fully vaccinated, 54.3% one booster, 23.4% two boosters). We detected Spike-IgG in 95.6%, Nucleocapsid-IgG in 24.0%, and neutralization against Wu01, BA.4/5 and BQ.1.1 in 94.4%, 85.0%, and 73.8% of participants, respectively. Neutralization against BA.4/5 and BQ.1.1 was 5.6- and 23.4-fold lower compared to Wu01. Accuracy of S-IgG detection for determination of neutralizing activity against BQ.1.1 was reduced substantially. We explored previous vaccinations and infections as correlates of BQ.1.1 neutralization using multivariable and Bayesian network analyses. Given a rather moderate adherence to COVID-19 vaccination recommendations, this analysis highlights the need to improve vaccine-uptake to reduce the COVID-19 risk of immune evasive variants. The study was registered as clinical trial (DRKS00029414).

Population immunity against SARS-CoV-2 plays a key role in the course of the pandemic and determines morbidity and mortality of COVID-19[1–3]. To date, 3 years after the emergence of SARS-CoV-2, immune evasion presents the most significant challenge to combat COVID-19[4–8].

At the beginning of 2022, a rapid surge of infections was detected worldwide and driven by the Omicron variant BA.1 that exhibited substantial immune evasion properties[9–11]. Subsequently, multiple Omicron sub-lineages emerged, including BA.5, which accumulated additional mutations in the Spike protein and became the

predominant variant globally in June 2022[12]. To that date, BA.5 demonstrated the strongest immune escape from antibodies induced by either SARS-CoV-2 vaccination or infection as well as from therapeutic monoclonal antibodies[13–17]. However, the continuous evolution of SARS-CoV-2 gave rise to further sub-lineages, including BQ.1 and BQ.1.1, with a relative share of all sequenced variants worldwide of 0.1% in August but 49.7% in November 2022[18,19]. This increase was likely caused by additional immune evasion properties in the BQ.1 and BQ.1.1 subvariants, enabling infections of SARS-CoV-2 vaccinated and convalescent individuals. Indeed, early data on the neutralization resistance of BQ.1 and BQ.1.1 show that it is mainly driven by an N460K mutation while the R346T and K444T mutations of BQ.1.1 contribute to a lesser extent[20–23]. Given the immune evasive properties of BA.5 and BQ.1.1 variants, seroprevalence studies that include analyses on neutralizing activity are critical to assess COVID-19 immunity.

In this multicenter-study, we determined SARS-CoV-2 Spike (S)-IgG levels, Nucleocapsid (NC)-IgG levels, and serum neutralization against Wu01, BA.4/5 and BQ.1.1 in 1411 patients that received medical treatment at emergency departments of five university hospitals (maximum care hospitals) in North Rhine–Westphalia, Germany, between August and September 2022. We analyzed IgG levels and neutralization activity together with detailed information on the medical history and SARS-CoV-2 immune status of the participants. Finally, we conducted multivariable and Bayesian network analyses to provide a better understanding of factors associated with the quantity and quality of the antibody response to SARS-CoV-2. Our results inform on SARS-CoV-2 immune status and its predictive factors, which helps to assess COVID-19 risk in highly vulnerable groups.

## Results

### Characteristics of study participants to determine SARS-CoV-2 humoral immunity

During the time of sample collection, 10,191 patients sought medical treatment in the emergency departments. Of those patients, 1411 (13.9%) were enrolled for study participation. There was no significant difference between the age and sex distributions of all emergency department patients and the study participants, indicating the representativeness of the study sample for the source population (Supplementary Fig. 1, Supplementary Fig. 2). We collected serum samples from all participants and determined S- and NC-IgG reactivity using chemiluminescence immunoassay (CLIA) and enzyme-linked immunosorbent assay (ELISA), respectively (Fig. 1a, b). Pseudovirus neutralization assays were performed to determine serum neutralizing activity against SARS-CoV-2 Wu01, BA.4/5, and BQ.1.1 variants (Fig. 1a). Information on epidemiological and clinical data as well as information on COVID-19 vaccination status and previous infections were collected in structured interviews and extracted from medical records. Enrolled participants had a median age of 53 years (range 18–98, IQR: 35–69) with an overall balanced sex distribution (48.5% female; 51.3% male; Fig. 1c, Supplementary Table 1). In total, 64.2% of the participants reported pre-conditions related most frequently to cardiovascular (52.3%) and neoplastic (24.4%) diseases (Fig. 1d). 13.6% of the participants reported drug immunosuppression at the time of sample collection (Fig. 1e). Overall, 94.4% of the participants reported having received at least one dose of an EU-approved vaccine (Fig. 1f), and 45.7% reported at least one previous SARS-CoV-2 infection. The addition of the number of reported received shots and reported infections indicated that 50.8% of the participants were exposed to at least four previous S-antigen contacts from either vaccination or infection (Fig. 1g). A full vaccination was defined as the administration of two sequential doses of Comirnaty, Spikevax, Vaxzevria, JCOVDEN, or Nuvaxovid, according to German COVID-19 vaccination recommendations. An interval of 3–6 weeks between the two doses was recommended. A booster vaccination was defined as a dose of Comirnaty or Spikevax administered after full vaccination. A single booster

≥6 months after the last dose was recommended for persons aged 18 years and older. Additionally, a second booster was recommended ≥6 months after the last dose for persons aged 60 years or older. Hybrid immunity was considered by the vaccination recommendation as infections could be counted as antigenic contacts and, by that, potentially substitute for recommended doses of vaccination. However, the substitution of vaccinations with infections was dependent on the time interval between sequential antigenic contacts and was considered a complex aspect of the guidelines. When stratified by age and taking previous infections into account, 67.7% of all participants were vaccinated according to German COVID-19 vaccination recommendations[24] (Fig. 1f).

### High S-IgG seroprevalence in patients visiting emergency departments in North Rhine-Westphalia, Germany

To determine SARS-CoV-2 humoral immunity, we first measured seroprevalence and levels of S-IgG in all participants. S-IgG could be detected in 95.6% of the participants. Of the 4.4% S-IgG-negative participants, 27.9% reported drug immunosuppression at the time of sampling. Of those reporting no drug immunosuppression, 31.8%, 29.5%, 6.8%, 22.7%, 6.8%, and 2.3% reported 0, 1, 2, 3, 4, or >4 previous S-antigen contacts, respectively (Fig. 2a). Thus, considering German COVID-19 vaccination recommendations[24], 86.9% of the seronegative participants were either immunosuppressed and/or insufficiently vaccinated. Next, we characterized S-IgG levels stratified by sex, age, preconditions, drug immunosuppression, and the number of S-antigen contacts (Fig. 2b). S-IgG levels were not significantly different between females and males (GeoMean 1697 versus 1836 BAU/ml, Mann–Whitney test: $p = 0.663$), different age groups (18–30, 31–60, and >60 years; 2003, 1612, and 1858 BAU/ml; Kruskal–Wallis test: $p = 0.357$) and participants with or without pre-conditions (GeoMean 1884 versus 1698 BAU/ml; Mann–Whitney test: $p = 0.538$) but lower in participants with drug immunosuppression compared to those without (GeoMean 1042 versus 1908 BAU/ml, Mann–Whitney test: $p = 0.0005$). Furthermore, S-IgG levels were higher in participants with a higher number of reported S-antigen contacts (number of infections and received vaccinations). The S-IgG levels ranged from 86 to 4450 BAU/ml for 0 to >4 reported S-antigen contacts (Kruskal–Wallis test: $p < 0.0001$). In addition, S-IgG levels were higher in those participants that reported more vaccinations or previous infections compared to those with a lower reported number and in those that tested NC-positive compared to those that tested NC-negative (Supplementary Fig. 3).

### NC-IgG seroprevalence and prediction of S-IgG levels

Next, we determined seroprevalence and levels of NC-IgG in all participants. NC-IgG could be detected in 24.0% of all enrolled participants (Fig. 2c). Of those participants, 77.3% reported previous SARS-CoV-2 infections. Conversely, 31.9% of participants who tested negative for NC-IgG reported previous SARS-CoV-2 infections. The fractions of NC-IgG-positive participants were higher in those that reported previous infections (10.0%, 39.0%, 61.8% for 0, 1, or 2 previous infections, respectively). Furthermore, NC-IgG values were higher in participants with 2 previous infections compared to those with 1 or no infection with mean Signal-to-Cutoff ratios (S/CO) of 2.5 and 2.4 for 0 and 1 infection versus 3.3 for 2 infections (Dunn's multiple comparisons tests: 0 vs. 1, $p = 0.038$; 0 vs. 2, $p = 0.029$). In all NC-IgG-positive or -borderline participants with 1 reported infection within the last 6 months ($n = 199$), mean S/CO decreased from 5.13 to 1.44 during the first 6 months after infection.

Given the observed association between different clinical and serological features and S-IgG levels, we performed multivariable analysis including sex, height, weight, BMI, age, pre-conditions, drug immunosuppression, number of reported vaccinations and infections, time since last vaccination and infection (months) and NC-IgG serostatus in a stepwise regression model to determine features

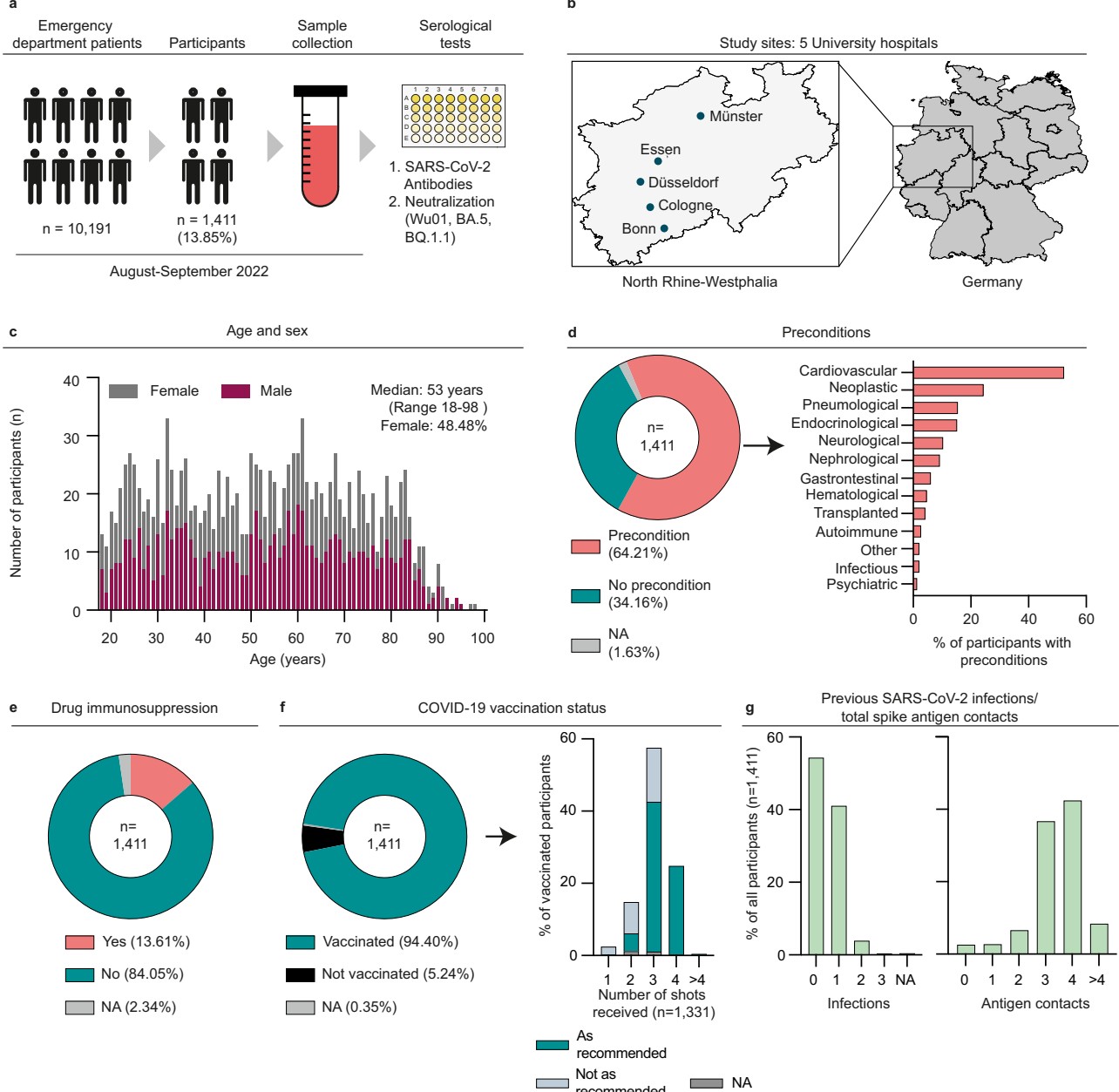

**Fig. 1 | Characteristics of study participants to determine SARS-CoV-2 humoral immunity. a** Illustration depicting number of study participants recruited in emergency departments, study timeline, and experimental procedures. **b** Illustration depicting the locations of the five study sites. Maps of Germany and North Rhine–Westphalia were designed with the iMapU tool provided by iExcelU. **c** Distribution of age and sex of the participants. **d** Pie chart depicting the presence of pre-conditions. Bar chart illustrating the distribution of pre-conditions stratified by organ system. **e** Pie chart depicting the presence of drug immunosuppression as reported by the participants at the time of sample collection. **f** Pie chart illustrating the vaccination status of the participants. Bar chart depicting reported vaccination scheme stratified by the number of received shots. Compliance with vaccination recommendations according to age is indicated by corresponding colors. **g** Bar charts illustrating reported previous infections and total Spike antigen contacts consisting of vaccinations and infections. Source data are provided as a Source Data file.

predicting S-IgG levels. Features were only added when they significantly improved the model according to a likelihood ratio test. This resulted in a ranking of the indicated predictiveness of all features according to their respective *p*-value during feature selection. NC-IgG, number of vaccinations, time since infection, time since last vaccination, drug immunosuppression, and number of infections were significant features for S-IgG prediction during feature selection. Sex, weight, BMI, height, pre-conditions, and age were no significant features for S-IgG prediction during feature selection (Fig. 3a). The final regression model (adjusted $R^2 = 0.241$) included only the significant features; we did not use the models' *p*-values, biased by the feature

selection, for subsequent interpretation. Bayesian network analysis revealed that NC-IgG serostatus, number of previous infections, and received vaccinations directly predicted S-IgG levels, while other parameters were only indirectly predictive (Fig. 3b).

**High S-IgG levels correlate with SARS-CoV-2 neutralizing activity**
To determine serum neutralization against Wu01 and BA.4/5 variants, we first determined the fraction of participants that showed detectable serum neutralization indicated by serum $ID_{50}s > 10$. Of all participants, 94.4% and 84.9% showed neutralizing activity against Wu01 and BA.4/5, respectively (Fig. 4a). The geometric mean $ID_{50}$ was

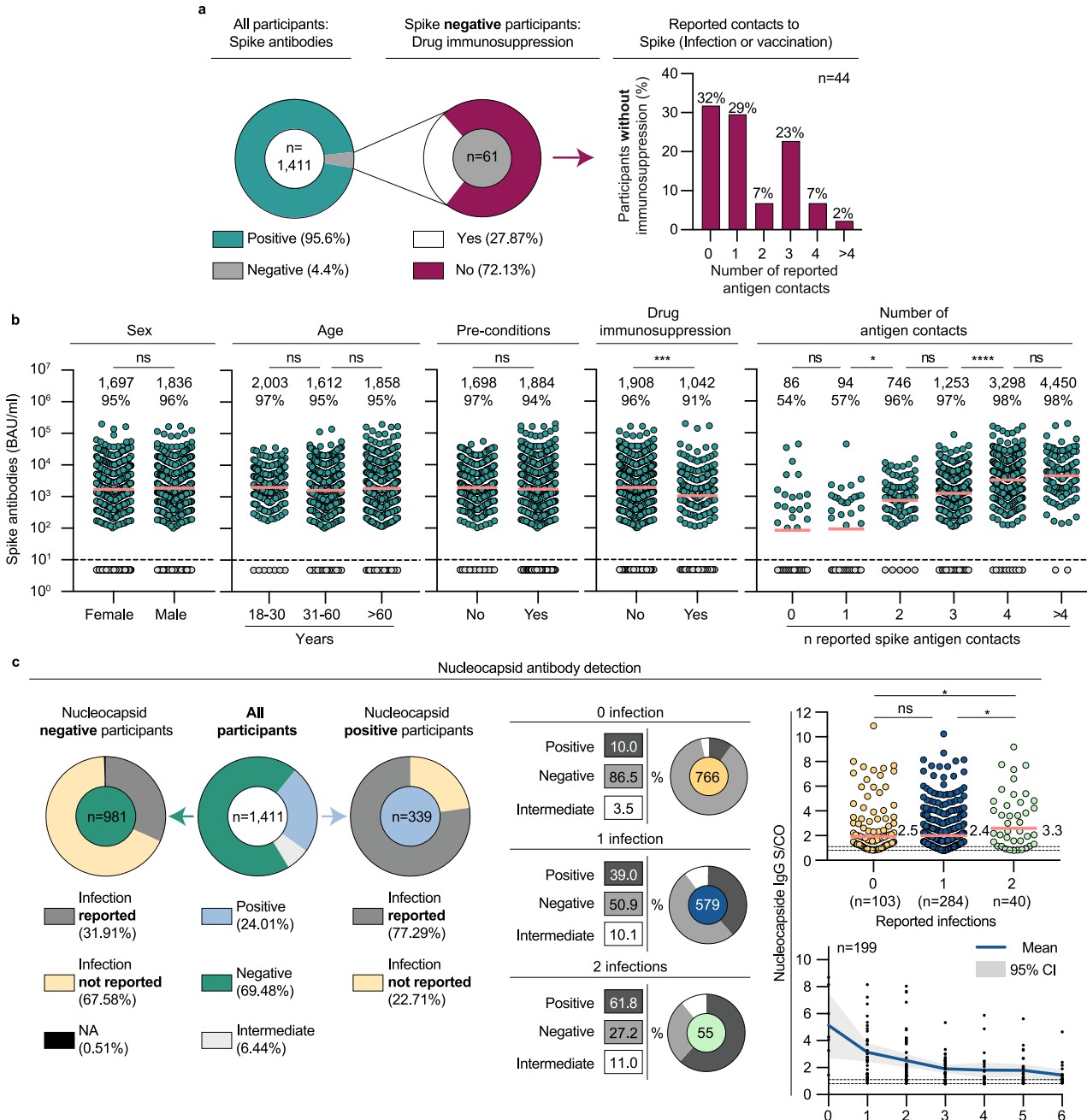

**Fig. 2 | Determination and description of the Spike- and Nucleocapside-IgG seroprevalence. a** Pie charts illustrating SARS-CoV-2 S-IgG prevalence and presence of drug immunosuppression of Spike-negative participants. Bar chart indicating the reported number of antigen contacts of participants with no detectable S-IgG and drug immunosuppression. **b** Dot plots depicting S-IgG BAU/ml values of all participants (*n* = 1411), subdivided based on sex, age, pre-conditions, drug immunosuppression, and a number of S-antigen contacts. The dotted lines represent the limit of detection (33.8 BAU/ml). Geometric means are indicated by horizontal red lines and listed in each plot over total fractions of participants with detectable S-IgG. Two-sided Mann–Whitney tests, Kruskal–Wallis-tests, and Dunn's multiple comparisons tests were performed for statistical analyses. Ns, *, **, ***, and **** represent *p*-values ≥ 0.05, <0.05, ≤0.01, ≤0.001, and ≤0.0001, respectively. **c** Left pie charts illustrate NC-IgG prevalence and fractions of participants

reporting any or reporting no previous infections in NC-IgG positive and negative participants, respectively. Right pie charts depict NC-IgG prevalence stratified by previous infections. The dot plot illustrates NC-IgG values (S/CO) stratified by the number of reported infections. The dotted lines represent the cut-off to negative and borderline values (0.8 and 1.1 S/CO). Means are indicated by horizontal red lines. Kruskal–Wallis and Dunn's multiple comparisons tests were performed for statistical analyses. * and ns represent *p*-values <0.05 and ≥0.05, respectively. *p*-Values were 0.317, 0.029, and 0.038, comparing 0 and 1, 0 and 2, and 1 and 2 previous infections, respectively. NC-IgG S/CO dynamic after infection is illustrated stratified by months after infection. Only data from individuals that reported one previous infection are shown. Individual values, mean S/CO, and 95% confidence interval are depicted by black dots, blue lines, or gray areas, respectively. Source data are provided as a Source Data file.

significantly lower for neutralizing activity against BA.4/5 than against Wu01 (243 vs. 1440) which corresponded to a 5.92-fold decrease in serum neutralization (Wilcoxon matched-pairs signed rank test: $p < 0.0001$; Fig. 4a). To explore a possible correlation between S-IgG levels and neutralizing activity, we next analyzed only those samples that showed detectable S-IgG and detectable serum neutralization against Wu01 and BA.4/5 ($n = 1176$). Of those samples, we performed Spearman correlation between S-IgG levels and $ID_{50}$ values. For

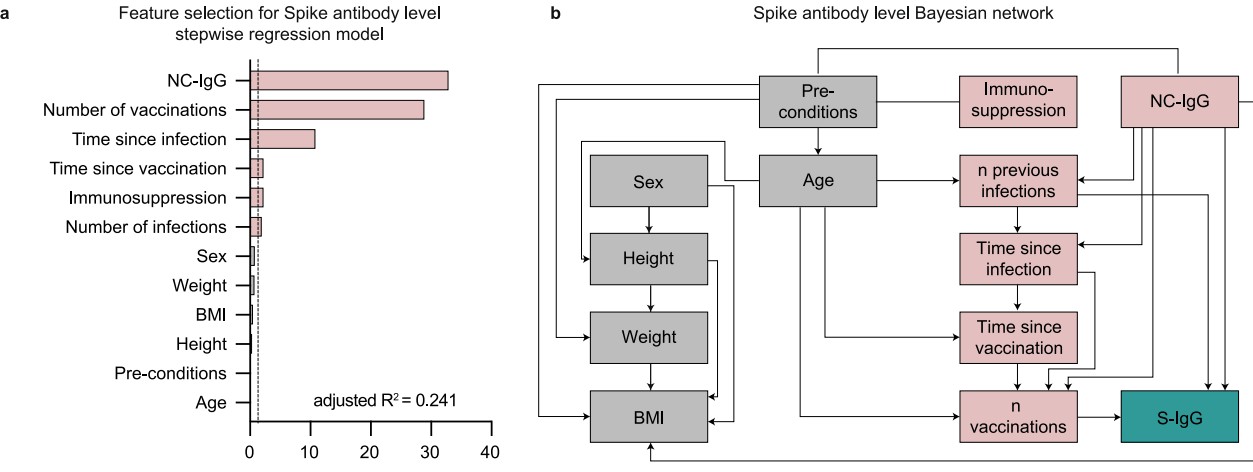

**Fig. 3 | Prediction of S-IgG levels. a** Stepwise forward regression model for predicting S-IgG (BAU/ml) using continuous features (age, height, weight, body mass index (BMI), number of infections, number of vaccinations, time since infection, and time since vaccination) and categorical features (sex, pre-conditions, NC-IgG, and immunosuppression). Features with $p < 0.05$ (two-sided likelihood ratio test) in the multivariable regression model are highlighted in red. **b** Bayesian network of the features predicting S-IgG. The graph connects the features, which are predictive of each other, with S-IgG as a sink. Features with $p < 0.05$ in the multivariable regression model (**a**) are highlighted in red.

neutralization against Wu01, the correlation coefficient was 0.74 ($p < 0.0001$), and for neutralization against BA.4/5, it was 0.62 ($p < 0.0001$; Fig. 4b). Further analysis of these samples revealed that in 32.4% and 64.7% of the participants with IgG-levels of 10–100 BAU/ml, no neutralizing activity could be detected against Wu01 and BA.4/5, respectively (Fig. 4c).

We conclude that IgG-detection can assess immunity against SARS-CoV-2. However, decreased accuracy for the prediction of serum-neutralizing activity introduced by immune evasive variants must be considered.

## S-IgG levels and S-antigen contacts contribute to serum neutralization against Wu01 and BA.4/5

To explore predictive features for serum neutralization, we first characterized neutralizing activity against Wu01 and BA.4/5 stratified by sex, age, drug immunosuppression, pre-conditions, number of received vaccinations, number of previous infections, and NC-IgG serostatus (Fig. 5). While serum neutralization was not significantly different between females and males, participants aged 18–30 years had a significantly higher geometric mean $ID_{50}$ compared to participants aged 31–60 and >60, both for neutralization against Wu01 and BA.4/5 ($ID_{50}$ GeoMean$_{Wu01}$: 2306 versus 1212 and 1405, Dunn's multiple comparisons tests: $p = 0.002$ and $p = 0.029$; $ID_{50}$ GeoMean$_{BA.4/5}$: 613 versus 231 and 167; Dunn's multiple comparisons tests: $p < 0.0001$ and $p < 0.0001$). Furthermore, serum neutralization was lower in participants that reported drug immunosuppression at the time of study participation in comparison to those that reported no drug immunosuppression ($ID_{50}$ GeoMean$_{Wu01}$: 1583 versus 747; $ID_{50}$ GeoMean$_{BA.4/5}$: 267 versus 117; Dunn's multiple comparisons tests: $p = 0.0006$ and $p = 0.002$). It was also lower in participants with pre-conditions compared to those without ($ID_{50}$ GeoMean$_{Wu01}$: 1919 versus 1227; $ID_{50}$ GeoMean$_{BA.4/5}$: 385 versus 190; Dunn's multiple comparisons tests: $p = 0.004$ and $p = 0.0001$). Serum neutralization against Wu01 and BA.4/5 was significantly higher in vaccinated participants in comparison to unvaccinated (Kruskal–Wallis test: $p < 0.0001$). In addition, serum neutralization was higher in participants that reported previous infections compared to those that did not (Kruskal–Wallis test: $p < 0.0001$) and in participants with detectable NC-IgG levels compared to those without (Mann–Whitney test: $p < 0.0001$) (Fig. 5).

In the following, we performed multivariable regression and Bayesian network analyses, as described above, to determine features

that contribute to serum neutralization (serving as a correlate of immune protection and disease severity of COVID-19)[25]. For neutralization against Wu01, multivariable regression (adjusted $R^2 = 0.5955$) showed S-IgG levels, time since infection, pre-conditions, time since vaccination, number of vaccinations, and number of infections to be predictive for neutralizing activity against Wu01. Bayesian network analysis revealed that S-IgG levels, NC-IgG serostatus, number of previous infections and vaccinations, and pre-conditions were directly predictive for serum neutralization against Wu01 (Fig. 6a, b). For serum neutralization against BA.4/5, the model (adjusted $R^2 = 0.5848$) showed S-IgG levels, number of previous infections, NC-IgG serostatus, age, and number of received vaccinations to be predictive for serum neutralization. Bayesian network analysis revealed that S-IgG levels, NC-IgG serostatus, age, number of previous infections, and time since infection were directly predictive for serum neutralization against BA.4/5 (Fig. 6a, b).

We concluded that the SARS-CoV-2 immune responses to Wu01 as well as BA.4/5 are determined by several factors. However, previous S-antigen contacts by both vaccinations and/or infections substantially contribute to SARS-CoV-2 humoral immunity.

## Impaired serum neutralization activity against Omicron sublineage BQ.1.1

After completion of sample collection, a rapid spread of the Omicron variant BQ.1.1 exhibiting three additional mutations in the Spike-protein in comparison to BA.4/5 (R346T, K444T, and N460K) could be observed (Fig. 7a, b). To determine serum neutralizing activity against BQ.1.1, we draw a proportionate stratified random sub-sample of 423 out of all 1411 participants (29.9%) to be additionally tested in pseudovirus neutralization assay against BQ.1.1. Strata were defined by sex and 10-years age categories (Supplementary Fig. 4a). There were no significant differences in age and sex distributions (Supplementary Fig. 4b) as well as in S-IgG levels or neutralizing activity against Wu01 and BA.4/5 (Supplementary Fig. 5) between the sub-sample and the entire study population. While neutralizing activity against Wu01 and BA.4/5 was detectable in 93.9% and 84.9% of the participants, respectively, only 73.8% of the studied participants presented activity against the Omicron sub-lineage BQ.1.1. Geometric mean $ID_{50}$s were 1302, 231, and 55 (Friedmann test: $p < 0.0001$; Dunn's multiple comparisons tests: $p < 0.0001$, $p < 0.0001$, $p < 0.0001$) for the respective variants. Overall, neutralizing activity against BQ.1.1 was 23.6-fold and 4.2-fold lower in

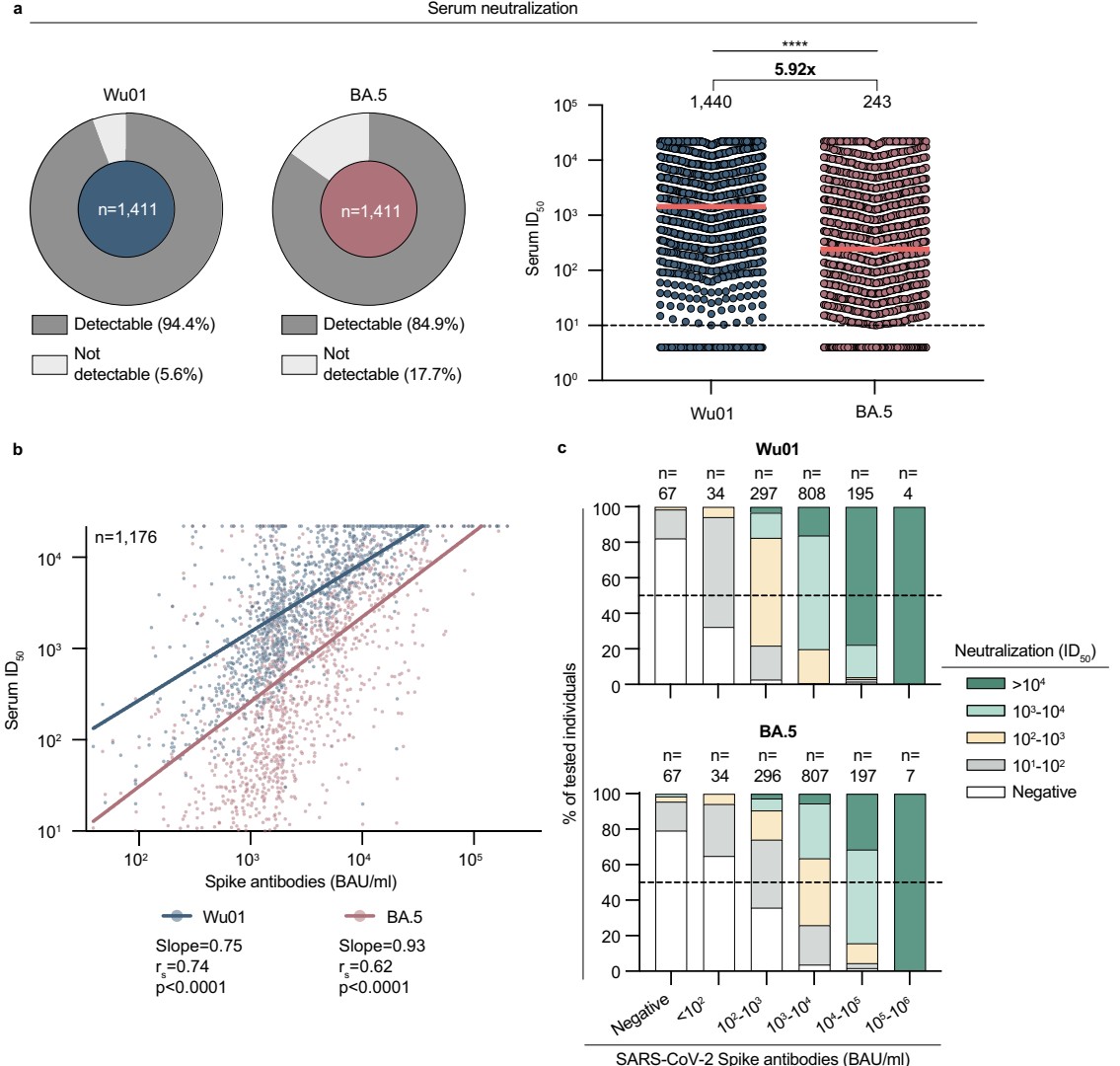

**Fig. 4 | High S-IgG levels correlate with SARS-CoV-2 neutralizing activity. a** Left: Pie charts illustrating SARS-CoV-2 serum neutralization prevalence of all participants (*n* = 1411; limit of detection ID$_{50}$ = 10). Right: Dot plot depicting serum ID$_{50}$ for Wu01 and BA.4/5 variants of all participants (*n* = 1411). The dotted lines represent the limit of detection (ID$_{50}$ = 10). Geometric means are indicated by horizontal red lines and listed for both variants over total fractions of participants with detectable serum neutralization. 5.92-fold decease is shown above data. A two-sided Wilcoxon matched-pairs signed rank test was performed for statistical analysis (*p* < 0.0001, represented by ****). **b** Spearman correlation of S-IgG (BAU/ml) against serum neutralization (ID$_{50}$) of Wu01 and BA.4/5 variants indicated by respective colors. Red and blue lines indicate fit lines computed with robust regression. Statistical significance was determined with a two-tailed *t*-test (*p* < 0.0001). **c** Serum neutralization (ID$_{50}$) is categorized as indicated and stratified by S-IgG (BAU/ml) for Wu01 und BA.4/5 variants. The dotted lines represent 50% of all participants in each stratum. *N* of each stratum is indicated on top of the graphs. Source data are provided as a Source Data file.

comparison to neutralizing activity against Wu01 and BA.4/5, respectively (Fig. 7c). The subsequent correlation of S-IgG values of all participants with detectable S-IgG against ID$_{50}$ values of all participants with detectable serum neutralization (*n* = 294) against Wu01 ($r_s$ = 0.64, *p* < 0.0001), BA.4/5 ($r_s$ = 0.44, *p* < 0.0001), and BQ.1.1 ($r_s$ = 0.48, *p* < 0.0001) revealed rather parallel fit lines and in comparison to Wu01 and a decrease of the intercepts for BA.4/5 and BQ.1.1, respectively (Fig. 7d, left panel). For S-IgG levels of 10–100 BAU/ml in 85.7%, and for S-IgG levels of 100–1000 BAU/ml in 55.6% of the participants, no serum neutralization against BQ.1.1 could be detected (Fig. 7d, right panel). Spearman correlations between ID$_{50}$ values of Wu01 versus BA.4/5 ($r_s$ = 0.75), Wu01 versus BQ.1.1 ($r_s$ = 0.67), and BA.4/5 versus BQ.1.1 ($r_s$ = 0.78) revealed a stronger correlation between Wu01 and BA.4/5 serum neutralization than between Wu01 and BQ.1.1 serum neutralization, reflecting the improved immune escape of BQ.1.1 (Fig. 7e).

Finally, we performed multivariable regression and Bayesian network analysis as described above to further explore and predict serum neutralization against BQ.1.1. The resulting model (adjusted $R^2$ = 0.477) showed S-IgG levels, number of previous infections, NC-IgG serostatus, age, and BMI to be predictive for serum neutralization against BQ.1.1 (Fig. 8a). Bayesian network analysis revealed that S-IgG levels, NC-IgG serostatus, time since infection, and age were directly predictive for serum neutralization against BQ.1.1 (Fig. 8b).

We concluded that serum neutralization of the participants is impaired against BQ.1.1. Furthermore, previous S-antigen contacts are predictive for serum neutralizing activity against BQ.1.1, as observed for Wu01 and BA.4/5.

## Discussion

Given the substantial immune escape of SARS-CoV-2 variants for the assessment of population immunity, completing S-IgG seroprevalence with serum neutralizing activity is essential[26,27]. It can help to assess the COVID-19 risk and inform on public health measures, such as vaccination strategies for Omicron-adapted vaccines,

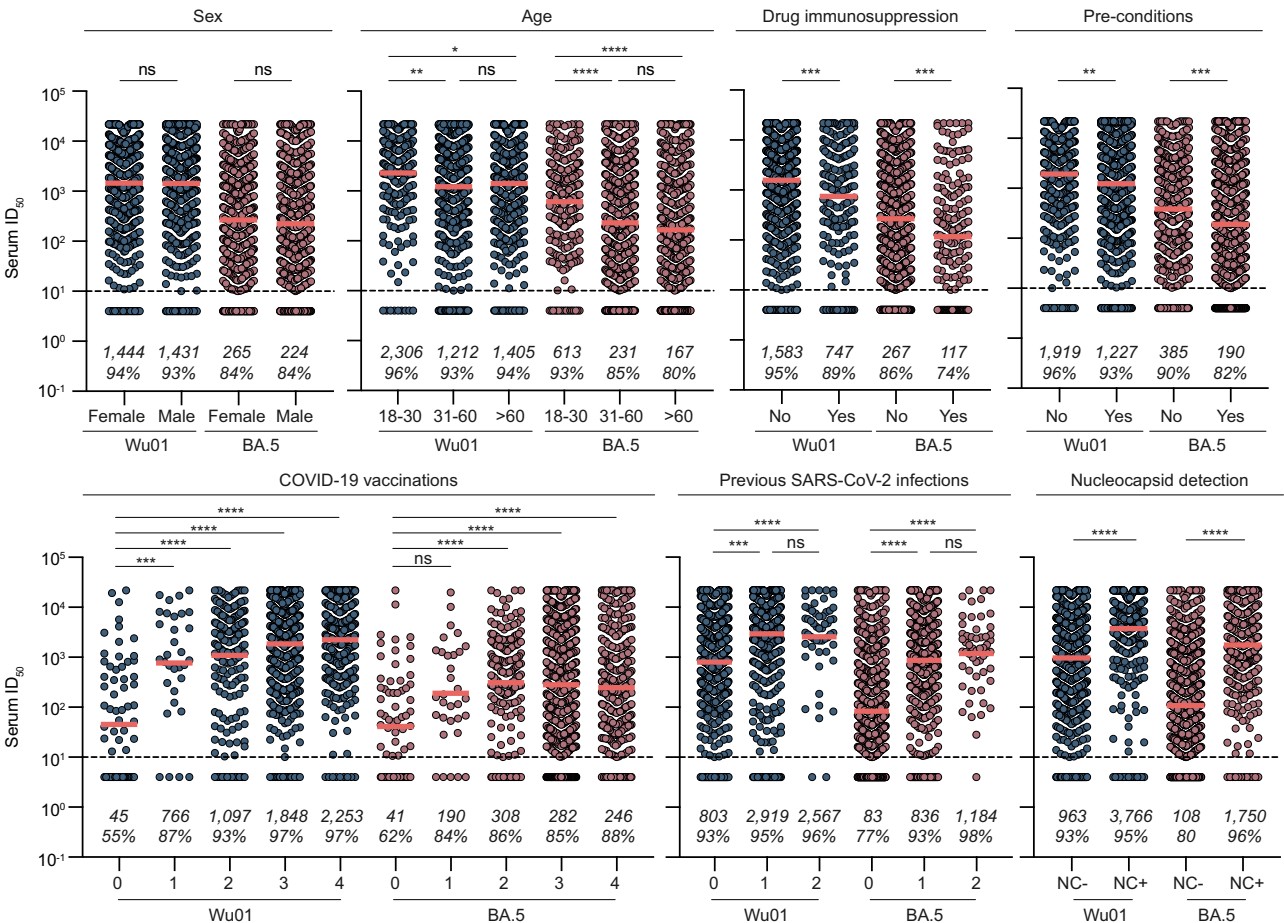

**Fig. 5 | Description of serum neutralizing activity.** Dot plots depicting serum neutralization ID$_{50}$ values for Wu01 (blue) and BA.4/5 (red), subdivided based on sex, age, drug immunosuppression, pre-conditions, number of vaccinations, number of infections, and NC-IgG detection of all participants ($n = 1411$). The dotted lines represent the limit of detection (ID$_{50} = 10$). Geometric means are indicated by horizontal red lines and listed in each plot over total fractions of participants with detectable serum neutralization. Two-sided Kruskal–Wallis tests, Dunn's multiple comparisons tests, and Mann–Whitney tests were performed for statistical analyses. Ns, *, **, ***, and **** represent $p$-values $\geq 0.05$, $<0.05$, $\leq 0.01$, $\leq 0.001$, and $\leq 0.0001$, respectively. Source data are provided as a Source Data file.

mask requirements, and further concepts of prevention. At the time of sample collection, there was an urgent need to assess the COVID-19 risk of vulnerable groups in autumn and winter of 2022/2023 in Germany. For that reason, we conducted a cross-sectional point-seroprevalence study in patients that received medical treatment in emergency departments.

Our results demonstrate a high fraction of participants (95%) with detectable S-IgG but only moderate adherence to current recommendations on COVID-19 vaccinations (67%)[24]. The fraction of participants that reported previous infections (45%) was high, and our results show that previous infections significantly contribute to neutralizing activity against SARS-CoV-2, which is in line with other studies[8,28–30]. Importantly, no data on the clinical course of the infections were available in our study. Accordingly, we do not conclude that infections rather than vaccinations represent a possible strategy in the future for boosting immunity in risk groups. Furthermore, in contrast to vaccinations, the timing and outcome of infections are not predictable.

Most strikingly, we showed a 23-fold decrease in serum neutralizing activity against BQ.1.1 in comparison to Wu01. This is in line with to-date limited data on BQ.1.1 immune escape and highlights BQ.1.1 as one of the variants with the greatest extent of immune escape that has been observed so far[13–17,22,23,31,32]. We could not differentiate the immune escape to infection-induced antibody response and/or antibodies induced by mono- or bivalent vaccinations. However, substantial neutralization resistance of BQ.1.1 after BA.5 infections were

shown previously[22], and bivalent vaccination was shown to elicit lower neutralizing activity against BQ.1.1 than against BA.5[21].

As shown in our multivariable regression model and Bayesian network analysis, S-IgG levels were predictive for neutralization activity against BQ.1.1. However, we find it important to emphasize that 59.6% of the individuals with detectable S-IgG < 1000 BAU/ml showed no detectable neutralizing activity against BQ.1.1, highlighting the decreased accuracy of S-IgG detection for assessing neutralization in individuals with low S-IgG titers. This information is relevant for clinical routine testing of S-IgG and needs to be considered when assessing COVID-19 risk in patients.

To the best of our knowledge, this study is one of the largest that assesses immune evasion of BA.4/5 and BQ.1.1 in a real-life setting. Furthermore, the comprehensiveness of data on medical history, vaccinations, and previous infections of the participants contribute to a reliable assessment of humoral immunity of vulnerable persons before the worldwide predominance of BQ.1.1. However, our study population is not a representative sample of the general population, and by that, the external validity of S-IgG seroprevalence is limited. Nevertheless, we expect that the observed impaired serum neutralization against BQ.1.1 can be generalized as it was controlled for several features including age, pre-conditions, drug immunosuppression, and a number of previous S-antigen contacts. Additionally, this study only analyzes immune evasive properties through neutralization assays. Thus, we cannot determine whether escape is mediated by impaired antibody binding or higher RBD-ACE2 (receptor binding

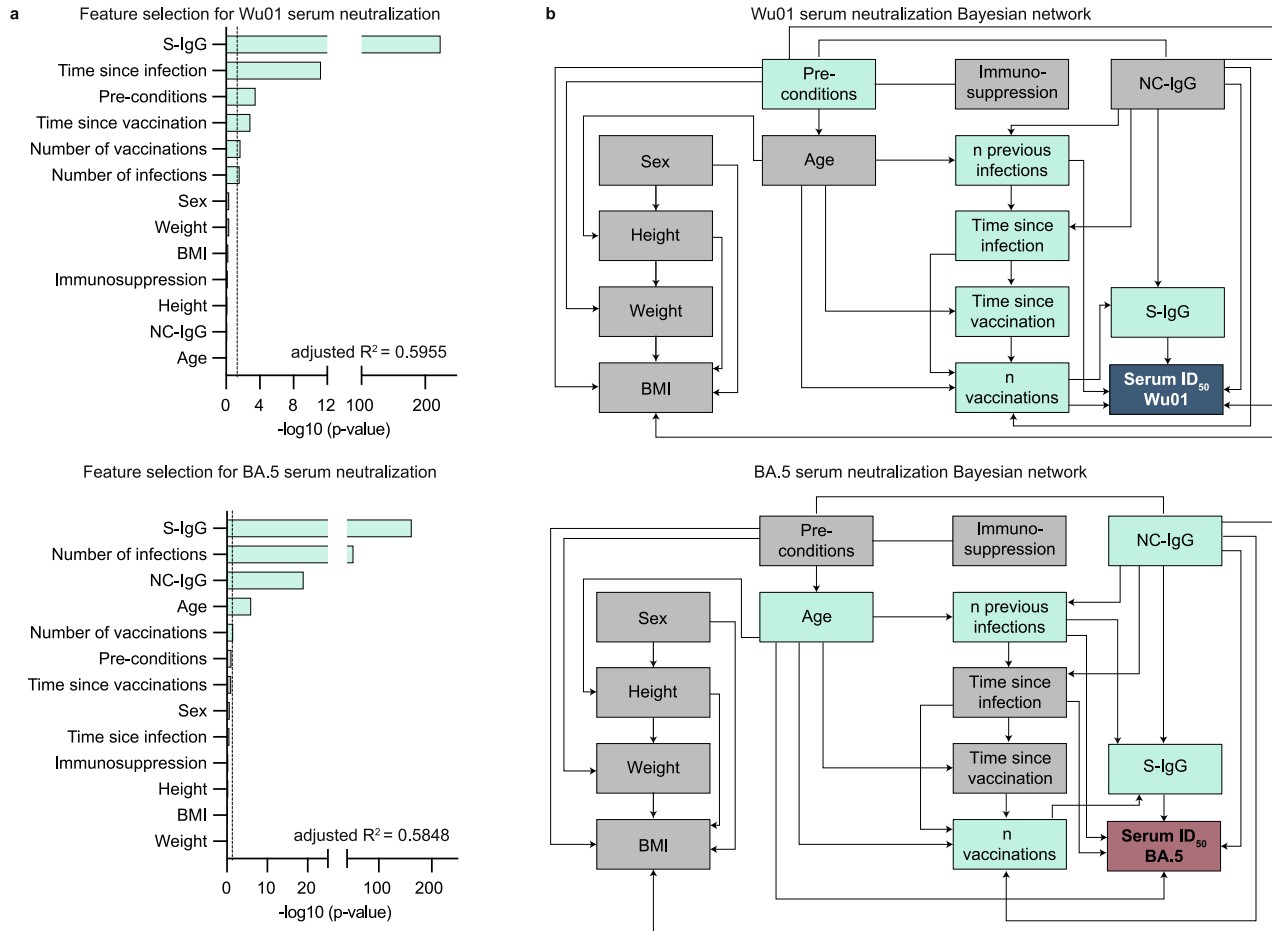

**Fig. 6 | S-IgG levels and S-antigen contacts contribute to serum neutralization against Wu01 and BA.4/5. a** Stepwise forward regression models for predicting serum neutralization ($ID_{50}$) using continuous features (age, height, weight, body mass index (BMI), number of infections, number of vaccinations, time since infection, time since vaccination, S-IgG) and categorical features (sex, pre-conditions, NC-IgG, and immunosuppression). Features with $p < 0.05$ (two-sided likelihood ratio test) in the multivariable regression model are highlighted in green. **b** Bayesian networks of the features predicting serum neutralization ($ID_{50}$). The graphs connect the features, which are predictive of each other with serum neutralization as a sink. Features with $p < 0.05$ (two-sided likelihood ratio test) in the multivariable regression model (**a**) are highlighted in green.

domain-angiotensin-converting enzyme 2) affinity[33,34]. Furthermore, we did not assess the Fc-dependent functions of the detected antibodies, which need to be considered when assessing the overall COVID-19 risk. It was previously described that vaccination induces antibodies that can leverage FcR binding across VOCs and that might contribute to a reduction in COVID-19 risk despite substantial immune escape[35,36]. Finally, our study does not assess whether inter-individual differences in the observed humoral immunity might be associated or casually linked with immune imprinting as described in detail previously[37,38].

In summary, we determined a high S-IgG seroprevalence, only moderate compliance with vaccination recommendations, and subsequently a broad range of serum neutralizing activity against BQ.1.1. By that, the observed substantial fraction of persons without detectable neutralizing activity mirrors the consequences of the interplay between immune escape and non-compliance with vaccination recommendations. We conclude that the improvement of vaccine uptake for all eligible individuals is critical for reducing the COVID-19 risk in upcoming waves of BQ.1.1 infections.

## Methods
### Ethical considerations
All samples and data were obtained under protocols approved by the ethics committees of the Medical Faculty of the University of Cologne (22_1262), of the Medical Faculty of the University of Bonn (314/22), of

the Medical Faculty of the University of Düsseldorf (2022-2072), of the Medical Faculty of the University of Essen (22-10838-BO), and of the Medical Faculty of the University of Münster (2022-490-b-S). All participants provided written informed consent. This study was registered as a clinical trial (DRKS00029414).

### Study design
Recruitment of participants and sample collection were conducted at five study sites in North Rhine-Westphalia, Germany (University Hospital of Cologne, University Hospital of Düsseldorf, University Hospital of Essen, University Hospital of Bonn, and University Hospital of Münster). Participation was offered to patients receiving medical treatment in emergency departments at one of the five study sites between August 8, 2022 and September 19, 2022. The study personnel recruited 1411 participants in cooperation with the emergency department personnel. Patients were required to meet the following eligibility criteria in order to be enrolled as participants: (i) only individuals aged ≥18 years were eligible, (ii) participants had to be patients at the emergency department at one of the five study sites, (iii) patients had to be able to consent to participate, (iv) the ability to consent was furthermore checked by the study personnel with a special focus on the medication, pain and the exceptional emotional situation of the patients, and (v) according to the assessment of the study personnel, participation in the study should not be a significant additional burden

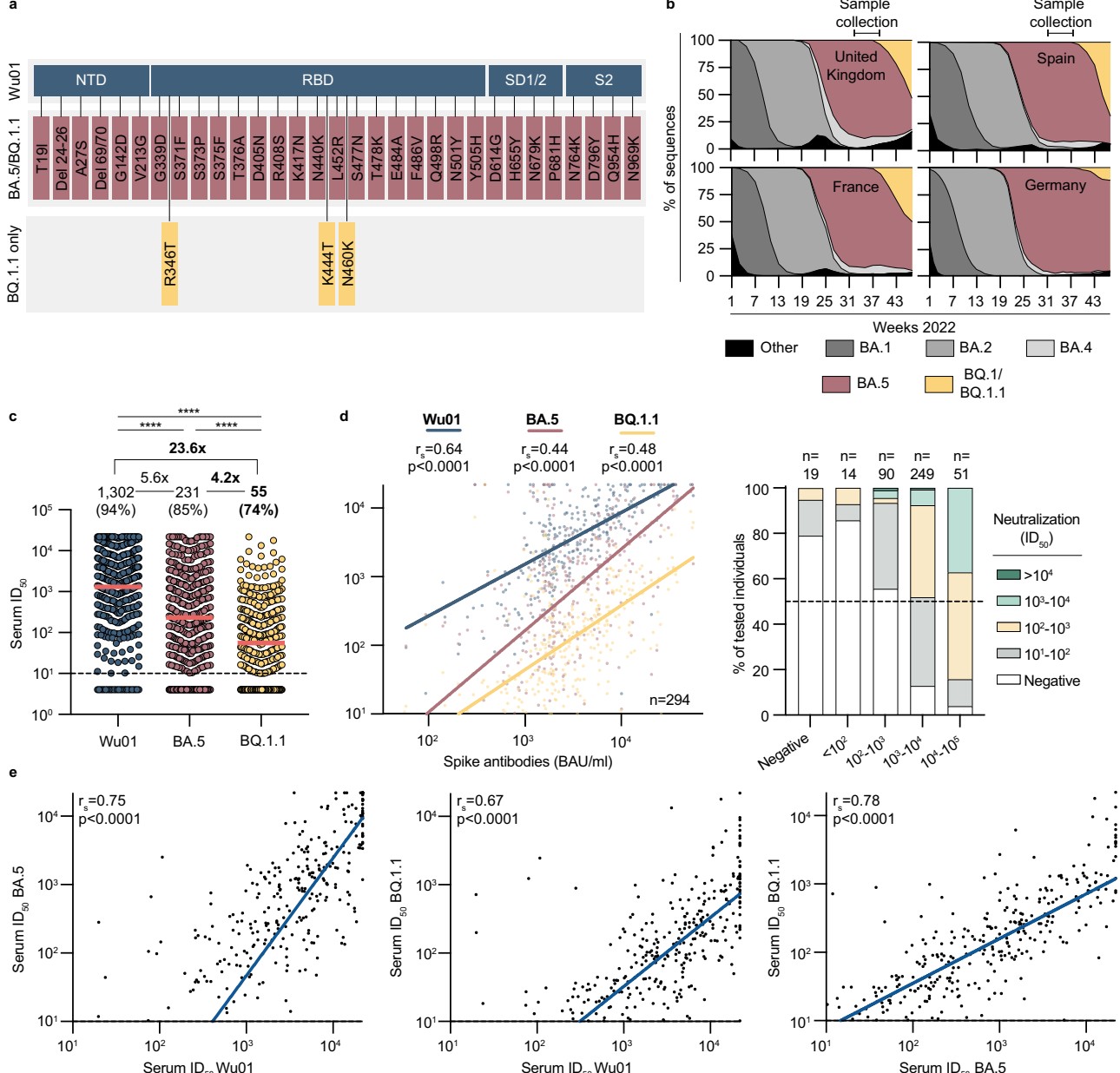

**Fig. 7 | Impaired serum neutralization activity against Omicron sub-lineage BQ.1.1. a** Spike amino acid changes in BQ.1.1 relative to BA.4/5 and Wu01. **b** Variant proportions in the United Kingdom, Spain, France, and Germany extrapolated from the bi-weekly Our World in Data dashboard and weekly reports of the Robert Koch Institute (accessed on November 25, 2022). NTD N-terminal domain, RBD receptor-binding domain, S Spike. **c** Dot plot depicting serum neutralization ID$_{50}$ values for Wu01 (blue), BA.4/5 (red), and BQ.1.1 (yellow) of the proportionate random sample of participants (n = 423). The dotted lines represent the limit of detection (ID$_{50}$ = 10). Geometric means are indicated by horizontal red lines and listed in each plot over total fractions of participants with detectable serum neutralization. X-fold decrease in serum neutralization is shown above data. Two-sided Friedman and Dunn's multiple comparison tests were performed for statistical analyses (all p-values were <0.0001, as indicated by ****). **d** Left: Spearman

correlation of Spike IgG (BAU/ml) against serum neutralization (ID$_{50}$) of Wu01, BA.4/5, and BQ.1.1 variants indicated by respective colors. Red, blue, and yellow lines indicate fit lines computed with robust regression. Right: Serum neutralization (ID$_{50}$) is categorized by indicated color and stratified by S-IgG (BAU/ml) for the BQ.1.1 variant. The dotted line represents 50% of all participants in each stratum. N of each stratum is indicated on top of the graph. Statistical significance was determined with a two-tailed t-test (all p-values were <0.0001). **e** Spearman correlations of Wu01 ID$_{50}$ against BA.4/5 ID$_{50}$ (left), Wu01 ID$_{50}$ against BQ.1.1 ID$_{50}$ (middle), and BA.4/5 ID$_{50}$ against BQ.1.1 ID$_{50}$ (right). Blue lines indicate fit lines computed with robust regression. Statistical significance was determined with a two-tailed t-test (all p-values were <0.0001). Source data are provided as a Source Data file.

for the patients. After verification of eligibility criteria and obtaining written informed consent by the study physicians, during blood collection, which was part of the standard medical treatment at the emergency departments, up to an additional 20 mL of blood was drawn from the participants to be analyzed in this study. Information on epidemiological data (age, sex, nationality, place of residence) and clinical data (body length and weight, pre-existing conditions,

immunosuppressive medication), as well as information on COVID-19 vaccination status (number of shots received, received vaccine type, date of vaccination), and past infections with SARS-CoV-2 (date of positive RT-qPCRs and/or rapid antigen detection tests) was verbally requested by the study personnel or extracted from the medical records of the participants. Samples and data collected at one study site were pseudonymized at that site and analyzed centrally at the

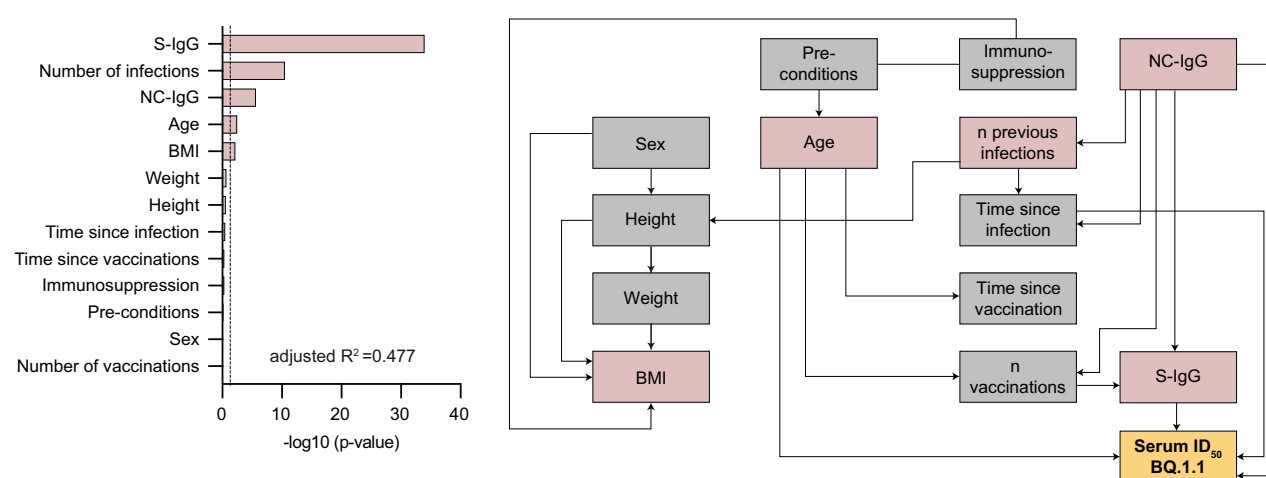

**Fig. 8 | Prediction of BQ.1.1 serum neutralizing activity. a** Stepwise forward regression model for predicting BQ.1.1 serum neutralization (ID$_{50}$) using continuous features (age, height, weight, body mass index (BMI), number of infections, number of vaccinations, time since infection, time since vaccination, S-IgG) and categorical features (sex, pre-conditions, NC-IgG, and immunosuppression).

Features with $p < 0.05$ (two-sided likelihood ratio test) in the multivariable regression model are highlighted in red. **b** Bayesian networks of the features predicting BQ.1.1 serum neutralization (ID$_{50}$). The graph connects the features, which are predictive of each other, with serum neutralization as a sink. Features with $p < 0.05$ in the multivariable regression model (**a**) are highlighted in red.

study site in Cologne. In addition, serum samples obtained at each study site were analyzed at that site to assess assay validity, as described further in "Methods Details".

### Processing of serum samples

Serum samples were collected in serum-gel tubes (Sarstedt) by venipuncture. After that, the samples were transported at 4 °C and processed within 48 h. After centrifugation, the serum was transferred to 2 ml cryotubes and stored at −80 °C till use. For further analysis in serological assays, the samples were thawed at room temperature, and 0.5 ml of each sample was pipetted into a 7 ml screwing tube (Sarstedt). For analysis in the pseudovirus neutralization assay, the samples were heat inactivated after thawing and processed as described below.

### Serological assays

Anti-SARS-CoV-2 antibodies were detected using commercial assays that use either SARS-CoV-2 Spike protein or SARS-CoV-2 Nucleocapsid antigens. All assays were used as per the manufacturer's recommendations.

For the main analysis of this study, S-IgG of all 1411 samples was measured using DiaSorin's LIAISON® SARS-CoV-2 TrimericS chemiluminescence immunoassay as described previously[39] with the following cut-off values: negative <33.8 BAU/ml and positive ≥33.8 BAU/ml. For validation of this assay, positive percent agreement (PPA) and negative percent agreement (NPA) with serological assays used at the other study sites were determined. PPA and NPA with (i) Anti-SARS-CoV-2 QuantiVac IgG BAU (Euroimmun) (*n* = 502, measured at the study site in Düsseldorf) were 99.78% and 83.33%, with (ii) DiaSorin's LIAISON® SARS-CoV-2 TrimericS chemiluminescence immunoassay (*n* = 185, measured at the study site in Essen) were 100% and 100%, with (iii) Abbott's anti-SARS-CoV-2 IgG Quant II chemiluminescence microparticle assay (Alinity i) (*n* = 133, measured at the study site in Bonn) were 100% and 66.66% (*n* = 3), and with (iv) Abbott's anti-SARS-CoV-2 IgG Quant II chemiluminescence microparticle assay (Alinity i) (*n* = 208, measured at the study site in Münster) were 99.5% and 100%. For graphical representation and statistical evaluation of serum samples, in Figs. 2, 4, 7, Figs. S3 and S5, samples that did not achieve IgG levels ≥33.8 BAU/ml were imputed to 4.81 BAU/ml (lower limit of quantification).

For the main analysis of this study, NC-IgG of all 1411 samples was measured using the Euroimmun anti-SARS-CoV-2-NCP-ELISA. Serum samples were tested on the automated system Euroimmun Analyzer I according to the manufacturer's recommendations. Results were indicated as Signal-to-Cutoff ratio (S/CO) values. That is, a cutoff value was established, and results were interpreted as a ratio to this cutoff value. S/CO values were interpreted as positive (S/CO ≥ 1.1), borderline (S/CO ≥ 0.8 < 1.1), and negative (S/CO < 0.8). For validation of this assay, positive percent agreement (PPA) and negative percent agreement (NPA) with serological assays used at the other study sites were determined. PPA and NPA with (i) Abbott's Architect SARS-CoV-2 IgG assay (*n* = 502, measured at the study site in Düsseldorf) were 97.6% and 94.85%, with (ii) Abbott's Architect SARS-CoV-2 IgG assay (*n* = 185, measured at the study site in Essen) were 100% and 95.95%, with (iii) Roche's Elecsys®-Assay (*n* = 133, measured at the study site in Bonn) were 56.86% and 100%, and with (iv) Abbott's Architect SARS-CoV-2 IgG (*n* = 208, measured at the study site in Münster) were 83.05% and 98.92%.

### Cell lines

HEK293T cells and 293T-ACE2 cells (Cat#CRL-11268 and Cat#NR-52511, respectively) were maintained in DMEM (Gibco) containing 10% FBS, 1% Penicillin-Streptomycin, 1 mM L-Glutamine and 1 mM Sodium pyruvate. Cells were grown in T75 flasks (Sarstedt) at 37 °C and 5% CO$_2$.

### Cloning of SARS-CoV-2 Omicron BA.4/5 and BQ.1.1 spike constructs

Cloning of Wu01- and BA.4/5 spike protein expression plasmids was previously described[13,40]. Compared with the Wu01 strain spike protein amino acid sequence, for the Omicron BA.4/5 strain, the following changes were included in the plasmid: T19I, Δ24-26, A27S, D69-70 G142D, V213G, G339D, S371F, S373P, S375F, T376A, D405N, R408S, K417N, N440K, L452R S477N, T478K, E484A, F486V Q498R, N501Y, Y505H, D614G, H655Y, N679K, P681H, N764K, D796Y, Q954H, and N969K mutations. For the BQ.1.1 spike protein expression plasmid, a gene fragment (Thermo Fisher) encompassing the additional R346T, K444T, and N460K mutations were cloned into the BA.4/5 spike protein expression plasmid using the NEB HiFi DNA Assembly Kit (New England Biolabs). All spike protein expression plasmids incorporate a C-terminal deletion of 21 cytoplasmic amino acids that results in

increased pseudovirus titers. Sanger sequencing was used for verification of the spike sequence.

## Pseudovirus neutralization assays

For pseudovirus particle production, HEK293T cells were co-transfected with plasmids encoding for the SARS-CoV-2 spike protein, HIV-1 Tat, HIV-1 Gag/Pol, HIV-1 Rev, and luciferase, followed by an internal ribosome entry site (IRES) and ZsGreen[41]. FuGENE 6 Transfection Reagent (Promega) was used for transfection. The virus culture medium was harvested 48–72 h after transfection and stored at −80 °C. The harvested virus was titrated. To this end, 293T-ACE2 cells[41] were infected and incubated for 48 h at 37 °C and 5% $CO_2$ prior to luciferase activity assessment. The activity was determined using a microplate reader (Berthold) after the addition of luciferin/lysis buffer (10 mM $MgCl_2$, 0.3 mM ATM, 0.5 mM Coenzyme A, 17 mM IGEPAL (all Sigma-Aldrich), and 1 mM D-Luciferin (GoldBio) in Tris-HCL). After heat inactivation (56 °C at 45 min), serum samples were serially diluted (1:3), starting with a 1:10 dilution. Before the addition of 293T-ACE2 cells, dilutions of serum samples were co-incubated with pseudovirus supernatants for 1 h at 37 °C. All samples were tested in single dilution series. Using the reagents described above, luciferase activity was determined after 48 h incubation at 37 °C and 5% $CO_2$. To quantify neutralization activity, after subtraction of background RLUs of non-infected cells, the 50% inhibitory dose ($ID_{50}$) was determined. $ID_{50}$ was defined as the serum dilution, which resulted in a 50% reduction in RLUs in comparison with the untreated virus control cells. GraphPad Prism 9 was used for the calculation of $ID_{50}$, which was plotted as dose–response curve. A SARS-CoV-2 neutralizing monoclonal antibody was used as run control (KV-Ab-188; R121-1F1, 1 μg/ml, 1:3 dilution series)[40]. Assay specificity was described before[42]. The average inter-assay coefficient was determined as 18.09% by testing 1546 serum samples in duplicates on different plates and on different days. For graphical representation and statistical evaluation of serum samples in Figs. 4, 5, and 7 and Fig. S5 samples that did not achieve 50% inhibition at the lowest tested dilution of 10 (lower limit of quantification, LLOQ) were imputed to $ID_{50} = 4$, and serum samples with $ID_{50}s > 21,870$ (upper limit of quantification) were imputed to $ID_{50} = 21,871$.

## Quantification and statistical analysis

We used stepwise forward regression to select features in a linear regression model. We distinguished between continuous features (age, height, weight, BMI, S-IgG, serum $ID_{50}$ Wu01, serum $ID_{50}$ BA.4/5, serum $ID_{50}$ BQ.1.1, number of previous infections, number of vaccinations, time since infection and time since vaccination) and categorical features (sex, preconditions, NC-IgG, and drug immunosuppression) (Figs. 3a, 6a and 8a). We removed all patient samples with no data for at least one feature, either missing or not specified. Our final data set had a size of 1209, 1209, 1207, and 352 for S-IgG, serum $ID_{50}$ Wu01, serum $ID_{50}$ BA.4/5, and serum $ID_{50}$ BQ.1.1, respectively. We transformed the features with a skewed distribution (S-IgG, Serum $ID_{50}$, and time since infection) by adding 1 and taking the log with base 10. We used stepwise forward regression for feature selection. We started with the base model, only including the intercept, and iteratively added the feature that increased the model fit the most. We tested the increase of the model fit with a likelihood-ratio test (R package lmtest). We did not include any Serum $ID_{50}$ features as predictors. We standardized the continuous features in our final model with mean zero and standard deviation 1 to allow for the comparison of the coefficients. For the next analysis, the larger data sets shrunk to 1181 participants because we included all features into one model, which led to more missing data in several participants. We created 1000 different datasets by sampling 1181 respectively 352 patients with replacement from the original participant dataset. We used the function for the t-distribution from the R-package stats (R Core Team 2022) to compute confidence intervals of model coefficients. To compute a Bayesian network for each dataset, we used the score-based hill-climbing algorithm (R package bnlearn) (Figs. 3b, 6b, and 8b). We restricted the response from having outgoing edges to preserve the directionality of the regression. We computed the edge fractions from the 1000 Bayesian networks and defined a consensus network by including edges with a fraction of 0.5, i.e., 500 or more appearances.

Testing for statistical significance of differences in sex distribution between the study population and the study sample was performed with the two-sided Fisher´s exact test. Testing for statistical significance of differences in S-IgG levels, NC-IgG S/CO values, or serum neutralization titers against Wu01, BA.4/5, and BQ.1.1 was performed with the two-sided Kruskal–Wallis test, Friedman test, Mann–Whitney $U$ test or Wilcoxon matched-pairs signed rank test, using Prism 9.0 (GraphPad). Dunn's multiple comparisons tests were performed as post hoc tests (GraphPad). Spearman's rank correlation coefficients ($R_s$) were determined using Prism 9.0 (GraphPad). Non-linear regression (robust regression) was calculated using Prism 9.0 (GraphPad). Statistical significance was defined as $p < 0.05$. Details are additionally provided in the Figure legends.

## Additional software

Maps of Germany and North Rhine-Westphalia were designed with the iMapU tool provided by iExcelU. Data collection was performed using Microsoft Excel for Mac (v.14.7.3.). Data analysis and Figure preparation was performed using Prism 9.0 (GraphPad). Data analysis was performed using RStudio (lmtest: 0.9-40, bnlearn: 4.8.1 stats: 4.2.2).

## Reporting summary

Further information on research design is available in the Nature Portfolio Reporting Summary linked to this article.

## Data availability

The generated data are available in the source data file. Raw data reported in this paper will be shared by the lead contact upon reasonable request. SARS-CoV-2 variant proportions were extrapolated from the bi-weekly. Our world in Data dashboard (http://ourworldindata.org, accessed on November 25, 2022) and weekly reports of the Robert Koch Institute. Source data are provided in this paper.

## Code availability

All original code has been deposited online[43] and is publicly available as of the date of publication.

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

## Acknowledgements

We are grateful to all study participants for their dedication to our research. We thank all members of all involved institutes. In particular, we thank Andrea Klöckner, Christian Overath, Anna Lucia Petrucci, Flaurant Ramadani, Claudia Reddig, Jasmin Lange, and Monika Eschbach-Bludau for processing serum samples and sample logistics. We thank Kanika Vanshylla for establishing the pseudovirus neutralization assay. We thank Thorsten Noack-Schönborn for providing data on emergency department patient characteristics. We thank Carsten Tschirner (IExcelU) for supporting data visualization and all laboratories providing SARS-CoV-2 sequencing information to GISAID and, subsequently, Our World in Data for facilitating variant distribution analyses. We acknowledge support for the Article Processing Charge from the DFG (German Research Foundation, 491454339). Funding was provided by the ministry for work, health, and social affairs of the state of North Rhine-Westphalia (CPS-1-1F).

## Author contributions

Conceptualization: F.D., F.K., J.T., S.L., H.S., C.E., M. Paluschinski, B.S., U.D.; J. Kü; Methodology: F.D., M. Pirkl., F.K.; Investigation: F.D., M. Paluschinski, F.K., E.A. B.M., V.D.C, V.B., H.G., F.T., M.B., M.M., M.L., M.A., M.T.H., W.H., M.M.M., P.K., J. Kü, M.S., G.O., J.K., C.E., C.K., D.E., I.G., M.K., W.O.M.-P., J.R., A.B., M.B., R.K.M., M.W., U.V.F., E.R.; Resources: F.K., J.T., H.S. U.D., S.L.; Formal analysis: F.D., M. Pirkl., and F.K.; Visualization: F.D.; Writing—original draft, F.D., M. Pirkl, and F.K.; Writing—review and editing, all authors; Supervision: F.K.; Funding acquisition: F.K., J.T., H.S. U.D., and S.L.

## Funding

## Competing interests

The authors declare no competing interests.

## Additional information

[1]Institute of Virology, Faculty of Medicine and University Hospital Cologne, University of Cologne, 50931 Cologne, Germany. [2]Institute of Virology, University Hospital Düsseldorf, Heinrich Heine University Düsseldorf, 40225 Düsseldorf, Germany. [3]Institute of Virology, Faculty of Medicine and University Hospital Muenster, University of Muenster, 48149 Muenster, Germany. [4]Institute for Virology, University Hospital Essen, University Duisburg-Essen, 45141 Essen, Germany. [5]Institute of Virology, University Hospital Bonn, University of Bonn, 53127 Bonn, Germany. [6]German Center for Infection Research (DZIF), Partner site Bonn-Cologne, 38124 Braunschweig, Germany. [7]Emergency Department, Medical Faculty and University Hospital of Düsseldorf, Heinrich Heine University Düsseldorf, 40225 Düsseldorf, Germany. [8]Division of General Internal and Emergency Medicine, Nephrology, Hypertension and Rheumatology, Department of Medicine D, Faculty of Medicine and University Hospital Muenster, University of Muenster, 48149 Muenster, Germany. [9]Center of Emergency Medicine, University Hospital Essen, 45147 Essen, Germany. [10]Occupational Medicine Department, University Hospital Bonn, 53127 Bonn, Germany. [11]Emergency Department, University Hospital Bonn, University of Bonn, 53127 Bonn, Germany. [12]Department II of Internal Medicine: Nephrology, Rheumatology, Diabetes and General Internal Medicine, Faculty of Medicine and University Hospital Cologne University of Cologne, 50931 Cologne, Germany. [13]Emergency Department, Faculty of Medicine and University Hospital Cologne, University of Cologne, 50931 Cologne, Germany. [14]Center for Molecular Medicine Cologne (CMMC), University of Cologne, 50931 Cologne, Germany. ✉e-mail: florian.klein@uk-koeln.de

