## [Peer review file · Nature Communications]

REVIEWER COMMENTS

Reviewer #1 (Remarks to the Author):

Dewald et al analysed S- and N-specific IgG responses as well as neutralising responses in sera from 1,411 individuals who received medical treatment in several emergency departments (Germany) between August and September 2022. The authors determined predictive factors (number of S-antigen contacts, age, sex, pre-conditions, drug immunosuppression) leading to neutralising responses against SARS-CoV-2 Wuhan, Omicron BA.5 and Omicron BQ1.1. The authors found that Wuhan N-specific IgG levels, the number of vaccinations, the number of infections and time since infection/vaccination as well as immunosuppression affected Wuhan S-specific IgG levels. Neutralisation against BA.5 is lower compared to Wuhan strain and ID50 correlated less with Wuhan S-specific IgG levels. The number of previous infections and vaccinations as well as Wuhan S-specific IgG levels were shown to impact on Wuhan- and BA.5-specific neutralising responses. Finally, the authors found that neutralising ability against BQ.1.1 was significantly reduced especially compared to Wuhan strain but also BA.5 in 423 participants. Wuhan S-specific IgG levels and the number of infections affected the most BQ.1.1-specific neutralising responses. In addition, BQ.1.1-specific neutralising responses correlated less with Wuhan S-specific IgG levels.

It is relevant to analyse neutralising activity of antibodies generated by infection(s) and/or vaccination(s) against emerged SARS-CoV-2 variants. It is also essential to estimate how Wuhan-specific seroprevalence can help to predict neutralising responses against variants. This paper is well-written, clear and the figures are very comprehensive. The statistical tests look ok to me. For the models, I am not fully expert so I will let other reviewers comment this point. This paper confirms similar findings observed in previous studies. It is known that the number of vaccinations and/or breakthrough infections improve neutralising responses against variants through an increase in IgG levels. Some studies have previously analysed neutralising responses against BQ.1.1. (10.1016/j.cell.2022.12.018, 10.1038/s41591-022-02162-x, 10.1016/S1473-3099(22)00805-2.).

Major comments:

- 1) I think this paper could be stronger and more novel if the quality of antibodies had been analysed more in details (e.g. avidity, Fc-dependent functions). Do the authors have those data ?
- 2) It is known that imprinting induced by infection with different variants may play a role on neutralising responses (10.1126/science.abq1841). How did the authors consider this point in their analyses ?
- 3) How did the authors standardise the ELISA data obtained using different commercial serological assays ?

Minor comments:

- 1) Line 128-132: This sentence related to vaccine status is not very clear. Do the authors mean at least one dose ? Or rather 1 scheme of immunization (2 or 3 doses) ? This sentence should be clarified.
- 2) Line 133: maybe it would be great to explain in a few words the German recommendations (number of doses, age range for boosters etc.) as in Figure 1 legend, it is written “as recommended” and “not as recommended”.

Reviewer #2 (Remarks to the Author):

The manuscript by Dewald, et al. describes a multi-center study that assesses serum antibody and neutralization against SARS-CoV-2 emerging variants in North Rhone-Westphalia, Germany. The authors measured spike IgG, nucleocapsid IgG, and serum neutralization against Wu01, BA.4/5, and BQ.1.1. The authors use this data in a multivariable regression model and Bayesian network analysis to demonstrate that spike antibody levels are most predictive of neutralization of all three strains including BQ.1.1, though notably, almost 60% of individuals with detectable S-IgG <1000 BAU/mL showed no detectable neutralizing activity against BQ.1.1. Consistent with what has been seen so far, the authors demonstrate a substantial (23-fold) decrease in serum neutralization activity against BQ.1.1 compared to Wu01. Overall, the authors' work is well-written and scientifically sound. However, the manuscript can be improved with some minor changes.

1. In Line 117, the authors state: “We collected serum blood samples from all participants and determined S- and NC-IgG reactivity using chemiluminescence immunoassay (CLIA) and enzyme-linked immunosorbent assay (ELISA)”. This leads to ambiguity about which assay (or both) were used for detecting antibodies against which antigen. “, respectively” should be added at the end of the sentence to clarify this (or the sentence should be rewritten otherwise).
2. The “Processing of serum” samples sections of the Methods appears to only describe transport and storage conditions, not the processing of the serum samples.

Reviewer #3 (Remarks to the Author):

General comments:

The authors present an analysis of data on SARS-CoV-2 immunity in a sample of emergency care patients in Germany. Results illustrate greater immune escape of the BQ.1.1 variant, compared with BA.4/5 and Wu01. Additionally, correlates of improved BQ.1.1 neutralization are identified.

I had some difficulty following the results section of the paper for at least the following reasons. First, the motivation for performing the analyses that the authors presented is not always clear to me. While the results section provided several numerical summaries describing the data, the scientific relevance of the different analyses was often not apparent. To this end, I would recommend providing better motivation for the statistical analysis and focusing on reporting results that align with the main “take-home” messages of the article. I also think that the introduction and results section could possibly be tied together better. Second, there are grammatical issues throughout the manuscript, and many sentences should be re-worded for clarity.

I also recommend that the authors be careful about the language they use when describing statistical associations. For instance, in the abstract, the authors say that neutralization activity “was reduced”, rather than neutralization activity “was lower”. The use of the word “reduce” is confusing because: (1) it is not clear what independent variable is claimed to have the effect of reducing neutralization, and (2) “reduced” would only be appropriate if an effort was made to conduct formal causal inference. Similar language appears throughout the manuscript (e.g., misuse of the word “affect”), and I recommend that the authors check the appropriateness of their language.

Specific comments:

- Page 3: It would be important, to provide a more detailed description of the study population. At what time of year were patients admitted? How were patients sampled? Was this population at high risk of SARS-CoV-2 infection (e.g., many participants had multiple comorbidities)? I think it would be helpful to summarize some background information about the study population in one or two sentences.
- Page 3: In the abstract, I recommend using an alternative description of the sample’s vaccination status. I would guess that most readers would not know precisely what vaccinations were recommended to the participants, so the description provided could be made more informative to the reader. Perhaps it would be better to state the percentage of participants that received a full primary vaccine sequence, or the percentage that have also received a booster, if these data are available.
- Page 4: In the introduction, it would be helpful to better connect the discussion of immune escape with the study objectives. The relationship between the two could be made clearer.
- Page 6: Please state explicitly what German COVID-19 vaccine recommendations are.
- Page 7; Lines 152-156. This is somewhat difficult to read, and the wording is unclear. Perhaps the authors can rephrase for clarity.
- Page 7; Line 157: The use of “affect” here is not entirely appropriate because this is an observational study

- Page 7; Lines 161-163. Similarly, this sentence should also be rephrased, as it is currently difficult to understand.
- Page 8; Line 167: It would be good to define S/CO for unfamiliar readers
- Page 8; Lines 172-185: I have multiple issues with the authors' regression approach. First, stepwise selection is known to have major limitations. For instance, the resulting prediction model can be sub-optimal, and the inferences drawn from the final model can be invalid (i.e., the p-values are not correct because the same data are used both to construct a model and to test hypotheses based on the model). In addition, I would caution the authors against conflating "most predictive" and "most significant", as the phrases carry different meanings and are not interchangeable. A small p-value does not indicate that a variable is highly predictive. And given that the resulting R-squared is rather small, I think all of the predictors can, at best, be described as modestly predictive.
- Page 9; Lines 193-201: These sentences should be rewritten for clarity. It is fairly difficult for me to understand what exactly analysis was performed. For instance, I do not quite understand what subset of the sample is used in this analysis, and whether the same sample is used to estimate all reported correlation coefficients.
- Page 10; Line 216: The use of the word "affect" is not appropriate because this is an observational study
- Page 10; Line 227: Again, I caution the authors against use of "most predictive" here, as in my comments for page 8 above

REVIEWER COMMENTS

Reviewer #1 (Remarks to the Author):

Dewald et al analysed S- and N-specific IgG responses as well as neutralising responses in sera from 1,411 individuals who received medical treatment in several emergency departments (Germany) between August and September 2022. The authors determined predictive factors (number of S-antigen contacts, age, sex, pre-conditions, drug immunosuppression) leading to neutralising responses against SARS-CoV-2 Wuhan, Omicron BA.5 and Omicron BQ1.1. The authors found that Wuhan N-specific IgG levels, the number of vaccinations, the number of infections and time since infection/vaccination as well as immunosuppression affected Wuhan S-specific IgG levels. Neutralisation against BA.5 is lower compared to Wuhan strain and ID50 correlated less with Wuhan S-specific IgG levels. The number of previous infections and vaccinations as well as Wuhan S-specific IgG levels were shown to impact on Wuhan- and BA.5-specific neutralising responses. Finally, the authors found that neutralising ability against BQ.1.1 was significantly reduced especially compared to Wuhan strain but also BA.5 in 423 participants. Wuhan S-specific IgG levels and the number of infections affected the most BQ.1.1-specific neutralising responses. In addition, BQ.1.1-specific neutralising responses correlated less with Wuhan S-specific IgG levels.

It is relevant to analyse neutralising activity of antibodies generated by infection(s) and/or vaccination(s) against emerged SARS-CoV-2 variants. It is also essential to estimate how Wuhan-specific seroprevalence can help to predict neutralising responses against variants. This paper is well-written, clear and the figures are very comprehensive. The statistical tests look ok to me. For the models, I am not fully expert so I will let other reviewers comment this point. This paper confirms similar findings observed in previous studies. It is known that the number of vaccinations and/or breakthrough infections improve neutralising responses against variants through an increase in IgG levels. Some studies have previously analysed neutralising responses against BQ.1.1. (10.1016/j.cell.2022.12.018, 10.1038/s41591-022-02162-x, 10.1016/S1473-3099(22)00805-2.).

Response:

We thank the reviewer for the constructive and overall positive feedback.

Major comments:

1) *I think this paper could be stronger and more novel if the quality of antibodies had been analyzed more in details (e.g., avidity, Fc-dependent functions). Do the authors have those data?*

Response

We thank the reviewer for the helpful comment and agree that additional analyses on e.g., avidity or Fc-dependent functions could be interesting. However, in this manuscript we focused on binding and neutralizing activities and the suggested analyses were, unfortunately, beyond the scope of our work. Nevertheless, Fc-mediated functions, as part of SARS-CoV-2 immunity, were previously addressed by several publications including Kaplonek *et al.*, *Immunity* 2022 (PMID: 35090580) who showed that mRNA-vaccination is capable of inducing antibodies that can leverage FcR binding across VOCs while in comparison, antibodies induced by infection showed compromised FcR binding.¹ Furthermore, they showed that functional responses to VOCs, such as complement and neutrophil phagocytic functions, were more reduced after natural infection than after mRNA-vaccination. This is in line with a study by Richardson *et al.*, *Cell Host Microbe* 2022 (PMID: 35436444) that described that vaccinated individuals with Omicron BA.1 breakthrough infection showed significantly higher levels of antibody-dependent cellular phagocytosis as compared to unvaccinated individuals.² In addition, Bartsch *et al.* have recently shown in *Sci Transl Med* (PMID: 35289637) that COVID-19 vaccination induces antibodies with preserved FcR functions despite the substantial reduction in neutralizing activity against the Omicron variant.³ In summary, these data highlight the value of antibody functions that go beyond their neutralization and emphasize their potential to contribute to the reduction of COVID-19 risk. As we did not perform these analyses, we added this topic to the discussion of the revised manuscript in **lines 348-355**.

2) *It is known that imprinting induced by infection with different variants may play a role on neutralizing responses (10.1126/science.abq1841). How did the authors consider this point in their analyses?*

Response:

We agree with the reviewer that immune imprinting by infection with different variants may play a role on neutralizing responses of the participants. Thus, as recommended by the reviewer, we explored a possible effect of immune imprinting by previous infections with Wu01, Alpha,

or Delta variants in persons with Omicron breakthrough infections on neutralizing response to BA.4/5. We performed multivariable analysis that included the infection sequence as categorical variable and used BA.4/5 serum ID₅₀ as dependent variable (**Table 1**). The coefficients for the presence of the respective infection sequences were not statistically significant. Thus, this analysis does not indicate that the effect of immune imprinting is visible in our data set. However, as this specific analysis is limited by a relatively low number of persons meeting the requirements of the respective infection-sequences and subsequently by rather broad 95%-confidence intervals of the respective coefficients, we would rather not show these analyses in the manuscript.

Feature	Estimate	SE	p-value	95% CI	
				low	high
Intercept	0,50	0,09	<0,001	0,32	0,68
Age	0,05	0,03	0,08	-0,01	0,11
Female sex	-0,08	0,07	0,22	-0,21	0,05
Height	0,004	0,04	0,92	-0,07	0,07
Weight	-0,02	0,03	0,59	-0,07	0,04
BMI	0,003	0,02	0,91	-0,04	0,05
Pre-conditions	0,07	0,06	0,20	-0,04	0,19
Drug					
Immunosuppression	0,09	0,07	0,23	-0,06	0,23
Number of vaccinations	-0,005	0,03	0,86	-0,06	0,05
Months since last vaccination	0,01	0,03	0,69	-0,04	0,06
Months since last infection	-0,25	0,04	<0,001	-0,33	-0,17
S-IgG level (BAU/ml)	0,46	0,02	<0,001	0,41	0,51
Infection sequence: Wuhan - Omicron	0,15	0,20	0,45	-0,25	0,55
Infection sequence: Alpha - Omicron	-0,21	0,46	0,65	-1,12	0,70
Infection sequence: Delta - Omicron	-0,13	0,29	0,64	-0,69	0,43
Infection sequence: Omicron only	-0,92	0,09	<0,001	-1,09	-0,75

Table 1: Multiple linear regression to explore immune imprinting on neutralizing activity against BA.4/5 in participants with Omicron breakthrough infections

However, our group has recently finalized a work that investigated on immune imprinting. In this study, longitudinal analyses of the memory B cell response to repeated SARS-CoV-2 antigen exposure by Wu01 booster vaccination and Omicron breakthrough infection were performed. In contrast to booster vaccination, breakthrough infection substantially enhanced the SARS-CoV-2 spike specific memory B cell pool. However, this was not primarily

attributable to B cell maturation, but to a selection of preexisting memory B cells resulting in significantly higher fractions of neutralizing antibodies. Broadly reactive B cell clones arose early and even neutralized highly mutated variants like XBB.1.5 that the study participants had not been in contact with. These data show that SARS-CoV-2 immunity is largely imprinted on Wu01 over the course of multiple antigen contacts but can respond to new variants through preexisting diversity. For further reading of this study, we have provided this manuscript to the editor.

3) How did the authors standardize the ELISA data obtained using different commercial serological assays?

Response:

We thank the reviewer for this question and agree that standardizing of results from different commercial serological assays is critical for a reliable interpretation of results. To this end, our analysis of serological measurements in the results part of this manuscript is based on only one commercial serological assay for S-antibodies and NC-antibodies, respectively. All 1,411 samples were measured with the same instrument at the Institute for Virology in Cologne. For the detection of S-antibodies, we used DiaSorin's LIAISON® SARS-CoV-2 TrimericS chemiluminescence immunoassay and for the detection of NC-antibodies, we used Euroimmun anti-SARS-CoV-2-NCP-ELISA. Only the results from those measurements were used for the analysis.

Since our study is a multi-center study, samples were collected at 5 study sites. In addition to the analysis in Cologne as described above, each sample was additionally analyzed at its respective study site in different serological assays. We calculated the positive and negative percent agreement between the results of the serological assays from the other 4 study sites with those results from the assays used at the study site in Cologne to determine its validity. We clarified our procedure in **lines 397-400, 414-419, and 436-438**.

Minor comments:

1) *Line 128-132: This sentence related to vaccine status is not very clear. Do the authors mean at least one dose? Or rather 1 scheme of immunization (2 or 3 doses)? This sentence should be clarified.*

Response:

We agree with the reviewer that this sentence could be formulated clearer and revised it as shown in **lines 126-128**.

2) *Line 133: maybe it would be great to explain in a few words the German recommendations (number of doses, age range for boosters etc.) as in Figure 1 legend, it is written “as recommended” and “not as recommended”.*

Response:

We agree with the reviewer that the German recommendations need to be explained in more detail. We added additional information on the German recommendations in **lines 130-144**.

Reviewer #2 (Remarks to the Author):

The manuscript by Dewald, et al. describes a multi-center study that assesses serum antibody and neutralization against SARS-CoV-2 emerging variants in North Rhine-Westphalia, Germany. The authors measured spike IgG, nucleocapsid IgG, and serum neutralization against Wu01, BA.4/5, and BQ.1.1. The authors use this data in a multivariable regression model and Bayesian network analysis to demonstrate that spike antibody levels are most predictive of neutralization of all three strains including BQ.1.1, though notably, almost 60% of individuals with detectable S-IgG <1000 BAU/mL showed no detectable neutralizing activity against BQ.1.1. Consistent with what has been seen so far, the authors demonstrate a substantial (23-fold) decrease in serum neutralization activity against BQ.1.1 compared to Wu01. Overall, the authors' work is well-written and scientifically sound. However, the manuscript can be improved with some minor changes.

Response:

We thank the reviewer for the constructive and overall positive feedback and hope that the implementation of the suggested changes is appropriate.

1. In Line 117, the authors state: "We collected serum blood samples from all participants and determined S- and NC-IgG reactivity using chemiluminescence immunoassay (CLIA) and enzyme-linked immunosorbent assay (ELISA)". This leads to ambiguity about which assay (or both) were used for detecting antibodies against which antigen. ", respectively" should be added at the end of the sentence to clarify this (or the sentence should be rewritten otherwise).

Response:

We thank the reviewer for this comment and added ",respectively" at the end of the sentence (lines 114-116).

2. The “Processing of serum” samples sections of the Methods appears to only describe transport and storage conditions, not the processing of the serum samples.

Response:

We agree with the reviewer and described the processing of the serum samples in more detail in **lines 402-409**.

Reviewer #3 (Remarks to the Author):

General comments:

The authors present an analysis of data on SARS-CoV-2 immunity in a sample of emergency care patients in Germany. Results illustrate greater immune escape of the BQ.1.1 variant, compared with BA.4/5 and Wu01. Additionally, correlates of improved BQ.1.1 neutralization are identified.

I had some difficulty following the results section of the paper for at least the following reasons. First, the motivation for performing the analyses that the authors presented is not always clear to me. While the results section provided several numerical summaries describing the data, the scientific relevance of the different analyses was often not apparent. To this end, I would recommend providing better motivation for the statistical analysis and focusing on reporting results that align with the main “take-home” messages of the article. I also think that the introduction and results section could possibly be tied together better. Second, there are grammatical issues throughout the manuscript, and many sentences should be re-worded for clarity.

I also recommend that the authors be careful about the language they use when describing statistical associations. For instance, in the abstract, the authors say that neutralization activity “was reduced”, rather than neutralization activity “was lower”. The use of the word “reduce” is confusing because: (1) it is not clear what independent variable is claimed to have the effect of reducing neutralization, and (2) “reduced” would only be appropriate if an effort was made to conduct formal causal inference. Similar language appears throughout the manuscript (e.g.,

misuse of the word “affect”), and I recommend that the authors check the appropriateness of their language.

Response:

We thank the reviewer for the constructive feedback and first explained in more detail the motivation for the applied statistical analyses as indicated in **lines 189, 210-212, 244-247, and 292-293**. We hope that this makes the manuscript more understandable and strengthens the take-home messages. Furthermore, as suggested, we tied the introduction and the results part together to avoid repetitive parts of the description of the overall study design as indicated in **lines 94-114**. Secondly, we checked the manuscript for grammatical issues and implemented the suggested changes of the words “reduction” and “affecting” as mentioned below, to express the observations of the study more appropriately.

Specific comments:

• Page 3: It would be important, to provide a more detailed description of the study population. At what time of year were patients admitted? How were patients sampled? Was this population at high risk of SARS-CoV-2 infection (e.g., many participants had multiple comorbidities)? I think it would be helpful to summarize some background information about the study population in one or two sentences.

Response:

We agree with the reviewer and thank for this comment. A more detailed description of the study population was added to the abstract on **page 3 (lines 53-59)**.

- *Page 3: In the abstract, I recommend using an alternative description of the sample's vaccination status. I would guess that most readers would not know precisely what vaccinations were recommended to the participants, so the description provided could be made more informative to the reader. Perhaps it would be better to state the percentage of participants that received a full primary vaccine sequence, or the percentage that have also received a booster, if these data are available.*

Response:

We agree with the reviewer that a more detailed description of the vaccination status is helpful. Thus, we included this description in the abstract in **line 59**. Furthermore, the vaccination status of the sample can be found in **Supplementary Table 1**.

- *Page 4: In the introduction, it would be helpful to better connect the discussion of immune escape with the study objectives. The relationship between the two could be made clearer.*

Response:

We thank the reviewer for this comment and reformulated the introduction in order to connect the discussion of immune escape of BA.5 and BQ.1.1 with the study objectives in **lines 91-104**.

- *Page 6: Please state explicitly what German COVID-19 vaccine recommendations are.*

Response:

We agree with the reviewer that the German recommendations need to be explained in more detail. We added additional information on the German recommendations in **lines 130-144**.

• *Page 7; Lines 152-156. This is somewhat difficult to read, and the wording is unclear. Perhaps the authors can rephrase for clarity.*

Response:

We thank the reviewer for this comment and rephrased the sentence for clarity as indicated in **lines 162-169**.

• *Page 7; Line 157: The use of “affect” here is not entirely appropriate because this is an observational study*

Response:

We thank the reviewer for this comment and rephrased the sentence without the use of “affect” (**lines 166-169**).

• *Page 7; Lines 161-163. Similarly, this sentence should also be rephrased, as it is currently difficult to understand.*

Response:

We rephrased the sentence to make it more understandable (**lines 172-176**).

• *Page 8; Line 167: It would be good to define S/CO for unfamiliar readers*

Response:

To make the text more understandable to unfamiliar readers, we added the definition of S/CO to the mentioned sentence in **lines 179-180**. Furthermore, we explained the concept of S/CO measurements in more detail in the methods part in **lines 433-435**.

• Page 8; Lines 172-185: *I have multiple issues with the authors' regression approach. First, stepwise selection is known to have major limitations. For instance, the resulting prediction model can be sub-optimal, and the inferences drawn from the final model can be invalid (i.e., the p-values are not correct because the same data are used both to construct a model and to test hypotheses based on the model).*

Response:

We agree with the reviewer that inferences drawn stepwise regression can be invalid and caution must be applied when interpreting p-values derived from feature selection and/or from the final linear model. Thus, we decided to exclude the p-value interpretation of the final linear model because our main aim was to only identify predictors for S-IgG levels. To illustrate this, we focused on the final feature selection and the R^2 of the final model in the revised manuscript without mentioning the p-values (**lines 191-199, 247-250, 253-256, and 294-296**).

In addition, I would caution the authors against conflating “most predictive” and “most significant”, as the phrases carry different meanings and are not interchangeable. A small p-value does not indicate that a variable is highly predictive. And given that the resulting R-squared is rather small, I think all of the predictors can, at best, be described as modestly predictive.

Response:

We thank the reviewer for his/her comment. We aimed to rank features based on the p-value obtained from a likelihood ratio test during selection to provide an indication of the predictiveness of the features. In the revised manuscript, the p-values of the coefficients of the final model are not interpreted in terms of predictiveness. As mentioned above we do not mention the p-values of the coefficients of the final model in the revised manuscript (**lines 191-199, 247-250, 253-256, and 294-296**) and avoided the terms “most predictive” and “most significant”.

• *Page 9; Lines 193-201: These sentences should be rewritten for clarity. It is fairly difficult for me to understand what exactly analysis was performed. For instance, I do not quite understand what subset of the sample is used in this analysis, and whether the same sample is used to estimate all reported correlation coefficients.*

Response:

We thank the reviewer for this comment and rewrote the mentioned sentences (**lines 212-217**).

• *Page 10; Line 216: The use of the word “affect” is not appropriate because this is an observational study*

Response:

We thank the reviewer for this comment and rephrased the sentence without the use of “affect” (**lines 232-238**).

• *Page 10; Line 227: Again, I caution the authors against use of “most predictive” here, as in my comments for page 8 above*

Response:

We corrected the misleading use of “most predictive” in **lines 247-250**.

References

1. Kaplonek, P., Fischinger, S., Cizmeci, D., Bartsch, Y.C., Kang, J., Burke, J.S., Shin, S.A., Dayal, D., Martin, P., Mann, C., et al. (2022). mRNA-1273 vaccine-induced antibodies maintain Fc effector functions across SARS-CoV-2 variants of concern. *Immunity* 55, 355-365.e4. 10.1016/J.IMMUNI.2022.01.001.
2. Richardson, S.I., Madzorera, V.S., Spencer, H., Manamela, N.P., van der Mescht, M.A., Lambson, B.E., Oosthuysen, B., Ayres, F., Makhado, Z., Moyo-Gwete, T., et al. (2022). SARS-CoV-2 Omicron triggers cross-reactive neutralization and Fc effector functions in previously vaccinated, but not unvaccinated, individuals. *Cell Host Microbe* 30, 880. 10.1016/J.CHOM.2022.03.029.
3. Bartsch, Y.C., Tong, X., Kang, J., Avendano, M.J., Serrano, E.F., García-Salum, T., Pardo-Roa, C., Riquelme, A., Cai, Y., Renzi, I., et al. (2022). Omicron variant Spike-specific antibody binding and Fc activity are preserved in recipients of mRNA or inactivated COVID-19 vaccines. *Sci Transl Med* 14, 9243.

REVIEWERS' COMMENTS

Reviewer #1 (Remarks to the Author):

The manuscript has been improved and clarified based on reviewers' comments. The authors properly replied to my comments.

Reviewer #3 (Remarks to the Author):

I have no further comments and feel that my previous concerns have been addressed appropriately.